# Solar wind erosion of lunar regolith is suppressed by surface morphology and regolith properties

Johannes Brötzner [1,8] ✉, Herbert Biber [1,8], Paul Stefan Szabo [2], Noah Jäggi [3], Lea Fuchs[1], Andreas Nenning [4], Martina Fellinger [1], Gyula Nagy [1,5], Eduardo Pitthan[5], Daniel Primetzhofer [5], Andreas Mutzke[6], Richard Arthur Wilhelm [1], Peter Wurz [7], André Galli [7] & Friedrich Aumayr [1]

Important aspects concerning the origin and formation of the Moon's exosphere, its tenuous gas envelope, remain puzzling with uncertainties regarding the importance of different effects. Two competing processes — micrometeoroid impact vaporization and solar wind ion sputtering — are considered key contributors to the ejection of particles into the exosphere. Here we present direct, high-precision yield measurements of solar wind ion sputtering using real lunar samples (Apollo 16 sample 68501), combined with advanced 3D simulations of regolith erosion. We find solar wind sputter yields up to an order of magnitude lower than previously used in exosphere models. The difference is primarily due to the suppressive effects of surface morphology, in particular the roughness and high porosity of the lunar regolith. Our results provide critical, experimentally validated sputter yield estimates and address long-standing modeling uncertainties. These results are particularly timely in light of upcoming and ongoing missions, such as the Artemis program at the Moon or BepiColombo at Mercury, contributing essentially to our understanding of how the surfaces of rocky bodies in the solar system are altered.

In the harsh space environment, the surfaces of planetary bodies are exposed to a variety of influences that erode them and change their surface properties. Besides mechanisms like photon-stimulated desorption, thermal desorption and vaporization by micrometeoroid impacts, sputtering by solar wind ions is a process particularly important for objects without protective atmosphere or magnetic field, like the Moon. There, solar wind ion bombardment has been shown to be responsible for the formation of nanophase iron particles, the darkening and reddening of reflectance spectra and the amorphization of rims on mineral grains[1–4]. Besides these, the release of surface species leads to the formation of a tenuous, collisionless gas envelope, the exosphere. Due to the high ejection energies, ion sputtering is a process particularly interesting as a supply for high-energy particles in the lunar exosphere, alongside the competing process of micrometeoroid impact vaporization[5]. Due to much higher ejection energies than impact-vaporized ejecta, sputtering contributes essentially to neutral and ionized escape of lunar matter[6].

There have been ample observations of the lunar exosphere[7–11] as well as copious efforts in modeling the complex interplay between the above-mentioned effects during the formation of an exosphere[12–17]. The relative contribution of different effects such as ion-induced sputtering and micrometeoroid impact vaporization still remains unclear, to a large extent due to gaps in our knowledge on accurate physical input parameters. Improving our understanding at the Moon would especially be important to aide understand planetary surface alteration at other exploration goals of current interest, such as Mercury and Phobos[18,19].

In the case of ion sputtering, one of the characteristic quantities for describing ion-induced erosion of planetary surfaces is the sputter yield, i.e., the number of atoms ejected per incoming ion. Understanding the sputtering process under relevant conditions and quantifying the sputter yields has been an ongoing undertaking, spanning decades with the first investigations predating even NASA's Apollo missions[20,21]. Subsequently, many samples were investigated, ranging from metals to analog minerals of

[1]Institute of Applied Physics, TU Wien, Vienna, Austria. [2]Space Sciences Laboratory, University of California, Berkeley, CA, USA. [3]Laboratory for Astrophysics and Surface Physics, University of Virginia, Charlottesville, VA, USA. [4]Institute of Chemical Technologies and Analytics, TU Wien, Vienna, Austria. [5]Department of Physics and Astronomy, Uppsala University, Uppsala, Sweden. [6]Max Planck Institute for Plasma Physics, Greifswald, Germany. [7]Space Research and Planetary Sciences, Physics Institute, University of Bern, Bern, Switzerland. [8]These authors contributed equally: Johannes Brötzner, Herbert Biber. ✉e-mail: broetzner@iap.tuwien.ac.at

varying degrees of surface roughness[20,22–29]. Moreover, numerical simulation tools have been employed to study the sputtering process, most notably the SRIM package[30]. In ensuing investigations, it was studied how to best model ion-solid interactions using the binary collision approximation (BCA), and a large literature body exists from which the SDTrimSP code emerged as a better alternative to SRIM[22,23,27,31–35]. Models have also taken into account different surface morphologies, from simple analytical models of sputtering to full-fledged 3D BCA calculations, mostly dealing with reflection of neutral atoms from regolith structures[36–43].

All these efforts notwithstanding, it was recently reported that previous assumptions on sputter yields are inconsistent with MESSENGER MASCS observations of Mercury's exosphere, suggesting that sputter yields be lower than previously estimated[16]. Similarly, isotopic analyses of Apollo soils revealed that over geological timescales, micrometeoroid impact vaporization is a more effective loss mechanism compared to sputtering on the lunar surface[44].

In this work, we investigate the factors leading to the overestimation of solar wind ion sputter yields in previous models, aiming to provide an essentially improved description of the sputtering behavior of planetary surfaces under solar wind ion bombardment. Our study combines experimental and numerical approaches to quantify sputter yields from actual lunar material rather than typically used analog minerals, irradiated with H and He ions at solar wind velocities of $\approx 440$ km s$^{-1}$. In particular, we report laboratory sputter yield measurements for both flat and rough samples prepared from Apollo soil 68501 and irradiated by 1 keV amu$^{-1}$ H and He, and compare against the results of SRIM, SpuBase, and SDTrimSP models. Moreover, we model the effect of regolith porosity in addition to rough, but compact, surface morphologies – an effect that is currently not accessible experimentally. We demonstrate that the commonly used simulation codes overestimate sputter yields of flat samples by more than a factor of 2 for hydrogen and helium ions at solar wind energies, underscoring the need for experimental validation using real lunar samples. Additionally, we show that surface roughness and the high porosity of lunar regolith further reduce sputter yields. Our findings provide realistic sputter yield estimates for actual lunar regolith of $7.3 \times 10^{-3}$ atoms ion$^{-1}$ and $7.6 \times 10^{-2}$ atoms ion$^{-1}$ for H and He, respectively, which are up to an order of magnitude smaller than previous estimates. These values are largely independent from the ion incidence angle, i.e., the solar zenith angle, and thus valid over a wide range of lunar latitudes. They are furthermore robust against slight variations in sample composition and therefore representative for lunar geology beyond the specific Apollo sample 68501.

## Results

### Flat sample sputter yields

Laboratory sputter yields of flat samples produced via pulsed laser deposition from Apollo 16 sample 68501 (see section Sample characterization) are given in Fig. 1 alongside simulation data of common modeling approaches. The sputter yields are given as function of incidence angle $\alpha$ in atomic mass units per incident ion (left axis of both panels) and atoms per incident ion (right axis of both panels). While our quartz crystal microbalance technique (QCM, see section Experimental methods) directly measures the sputter yields as mass changes (left axes), simulations typically give them in atoms per ion. Sputtered atoms per ion is also usually the unit of choice for exosphere models[12], which is why we give both units wherever possible. Conversion of the experimental data to atoms per ion was carried out under the assumption that in the steady state, the elemental sputtered particle fluxes correspond to the bulk stoichiometry[22,45].

For the H case, it is clear that all modeling attempts overestimate the experimental results of this work, denoted by the filled circles. SRIM (dashed line) predicts the highest values throughout most incidence angles. SDTrimSP with its default parameter set (dashed-dotted line) is lower, with an exception in the narrow region around $\alpha = 80°$. The two lines resulting in the lowest total mass yields are SDTrimSP using the parameter set published by Szabo et al.[22] (solid) and SpuBase with an improved hybrid compound binding model (dashed). While for smaller incidence angles, the two give almost identical results, the SpuBase curve rises steeper beyond $\alpha \approx 70°$ and reproduces the experimental trend better. SpuBase, however, cannot be extended to three-dimensional studies. We therefore proceeded to use the model by Szabo et al.[22] also for the subsequent 3D investigations and quantified the discrepancy between the solid line in Fig. 1 and the experimental data to be a factor of 0.47. This scaling factor was determined by taking the point-wise ratio between numerical and experimental data points and subsequently averaging these fractions.

A similar situation is observed for He. However, in this case, a clear separation between SRIM and the SDTrimSP variants is observed, and SRIM gives the highest sputter yield values over all incidence angles. Once more, SpuBase is the SDTrimSP-based curve without any adaptation closest to the measured data. On the other hand, the approach by Szabo et al.[22] results in the lowest mass yields. In this case, a factor of 0.66 is necessary to

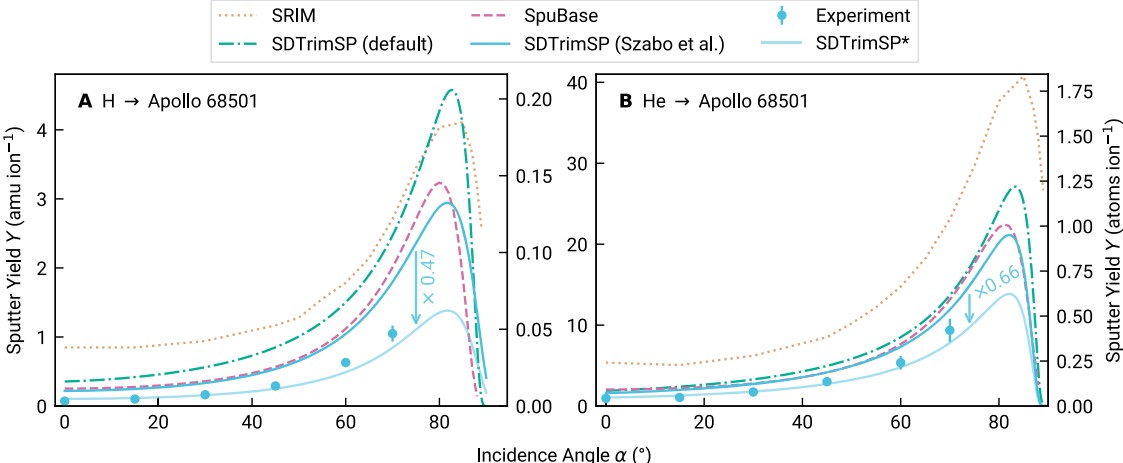

**Fig. 1 | Sputter yields $Y$ over incidence angle $\alpha$ for flat lunar samples under solar wind ion bombardment.** Comparison of various simulation models (lines) and experimental results (blue circles) for 1 keV amu$^{-1}$ H (**A**) and He (**B**) impactors, respectively. Simulation data stem from SRIM (dotted beige line), SpuBase (dashed, pink), SDTrimSP with the default parameters (dashed-dotted, teal) and with the parameters proposed by Szabo et al.[22] (solid, blue). Additionally, the arrows quantify the offset between experiment and simulation, and the lighter colored lines give the simulations scaled to match experimental data. The asterisk in "SDTrimSP*" denotes the use of the parameter set proposed by Szabo et al.[22] combined with this scaling factor. Experimental error bars are estimated from ion current fluctuations during the measurements and from the quality of fits to the QCM resonance frequency.

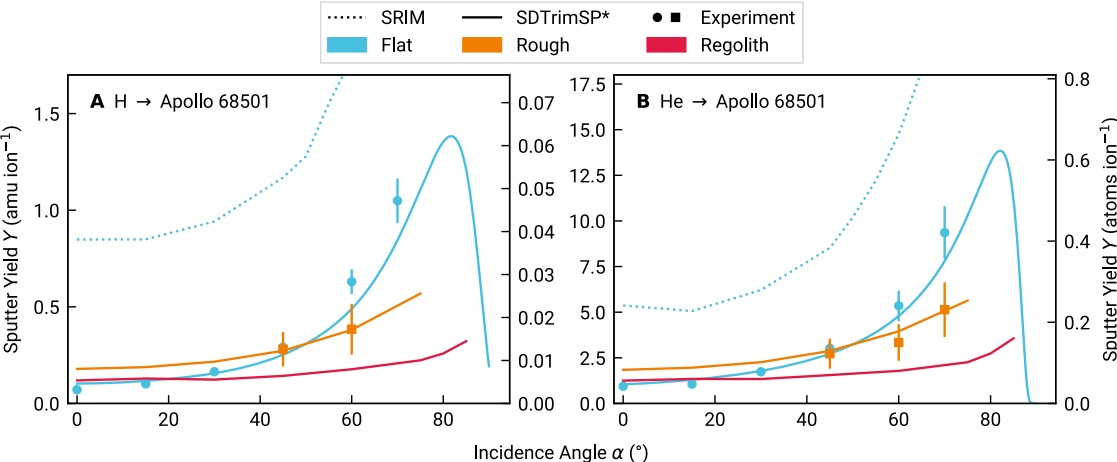

**Fig. 2 | Sputter yields $Y$ over incidence angle $\alpha$ of lunar surface material for typical solar wind impactors and different surface morphologies. A** Hydrogen impactors. **B** Helium impactors. Sputter yields are given for a flat surface (blue), a rough pressed pellet surface (orange) and a porous regolith structure (red). Experimental results are denoted by the symbols with error bars, while SDTrimSP-based and SRIM simulations are given by solid and dotted lines, respectively. Note that the scaling factor derived in Fig. 1 has been applied to the SDTrimSP simulations, as indicated by the labeling convention using the asterisk. Experimental error bars are estimated from ion current fluctuations during the measurements and from the quality of fits to the QCM resonance frequency.

match this curve to the experimental data points. While for He, the overestimation by SDTrimSP-based models is moderate, it is notably worse for H, the most prominent solar wind species, across all investigated numerical approaches.

### Surface morphology-dependent sputter yields

A comparison of the total sputter yields for flat, rough, and porous samples is given in Fig. 2A, B for 1 keV amu$^{-1}$ H and He, respectively. Just like in the previous subsection, experimental data for flat samples were obtained from films grown by pulsed laser deposition. For measurements on rough surfaces, we pressed some of the original regolith sample in order to form stable pellets that can withstand mounting in the vacuum vessel. Experimental data for these were obtained by means of a catcher QCM (cf. section Experimental methods). As this is not possible for the porous regolith case, only simulated data are available for this instance. For preparation and characterization, the reader is referred to section Sample characterization. Note that for SDTrimSP-based curves (both 1D and 3D), the projectile-dependent scaling factors of 0.47 and 0.66 (Fig. 1) are applied. Both ion species show a typical dependence of the sputter yield on the incidence angle when flat samples are considered (blue symbols and lines in Fig. 2). The yield increases with $\alpha$ until a maximum is reached for grazing incidence at roughly 80° for both He and H. Beyond this point, a sharp decrease is reported, and the sputter yield approaches 0 for near-horizontal incidence. Data for the rough pellet sample (orange lines) show a reduction in sputter yield for large incidence angles and an increase compared to the flat sputter yields (blue lines) for near-normal impact. The crossing point where sputter yields coincide between flat and rough samples is approximately located at $\alpha = 45°$ for both ion species.

As sputter yield measurements for loose regolith powder are not feasible in our setup, we introduced porosity in SDTrimSP-3D calculations (red lines). In this case, the same effect is observed in an even more pronounced way; the sputter yield is further flattened and reduced. Equal yields as compared to a flat surface are achieved near $\alpha = 20°$, but until this point, neither curve exhibits a particularly steep slope, such that also at normal incidence, the sputter yields are comparable. Beyond this region, the regolith yield stays almost constant, and only for grazing incidence above 75° to 80° a slight increase is discernible. When averaged over the simulated angle range, the regolith sputter yields for lunar soil are $7.3 \times 10^{-3}$ atoms ion$^{-1}$ for H and $7.6 \times 10^{-2}$ atoms ion$^{-1}$ for He. Because these reduced yields of porous regolith structures are largely independent of the ion incidence angle, they are applicable across a wide range of lunar latitudes.

### Discussion

BCA simulations have been used for decades and are still an active topic in research and development[46–48]. Nonetheless, common modeling approaches struggle with correctly describing sputter yields for compound materials (Fig. 1). At 45° incidence, the SRIM result for H is off by a factor of 8.2 from our regolith yields, which is particularly relevant as 45° yields were used to account for lunar surface roughness in previous exosphere models[12]. For more grazing incidences, the overestimation grows beyond an order of magnitude. It is thus evident that SRIM cannot reliably predict sputter yields for the lunar surface. This was to be expected, as the shortcomings of SRIM have been known and its use, particularly for energies in the solar wind relevant regime, is discouraged[49–53]. Also, the overestimation of the sputter yield by SRIM for mineral samples has been shown repeatedly[22,23,27]. Nevertheless, SRIM is still being used in recent literature[42,54,55]. As an alternative, SDTrimSP has been suggested, as its predictions are in better agreement with experimental results[22–24,27,56]. However, also for this code, parameter adaptations (or scaling) are necessary to reproduce sputter yields measured in experiments.

One possible shortcoming hindering a better understanding of the sputtering process in the BCA picture is the knowledge gap concerning surface binding energies (SBEs). The SBE directly influences the sputter yield[57,58] and is often approximated as the energy of sublimation[59,60]. This view, however, is debated, and subsequently, a substantial amount of research has been carried out on the physical meaning of the SBE and its importance for sputtering, also in the space sciences context[16,22,33,35,46,47,61]. As the SBE is strongly related to the energy spectra of sputtered ejecta[62], measurements of ejecta energy distributions could potentially clarify these matters and improve BCA simulations. However, few data are currently available for relevant compound materials, and only some data sets have been published for metallic samples or alkali halides[63–66]. For the time being, the model used in SpuBase[67,68] is the one studied BCA model closest to the experiment, where the binding energy approach is physically motivated and does not stem from fitting to a data set. Nonetheless, the remaining uncertainties in the BCA codes underline the necessity to validate simulated sputter yields with available experimentally measured data sets.

In addition to the overestimation by simulations, surface morphology further reduces the effective sputter yields (Fig. 2). A similar sputter yield reduction was found when comparing enstatite ($MgSiO_3$) thin film and pressed pellet samples and this effect was ascribed to surface roughness[24]. Indeed, a joint experimental and numerical study demonstrated that surface morphology is capable of lowering the sputter yield and highlighted a

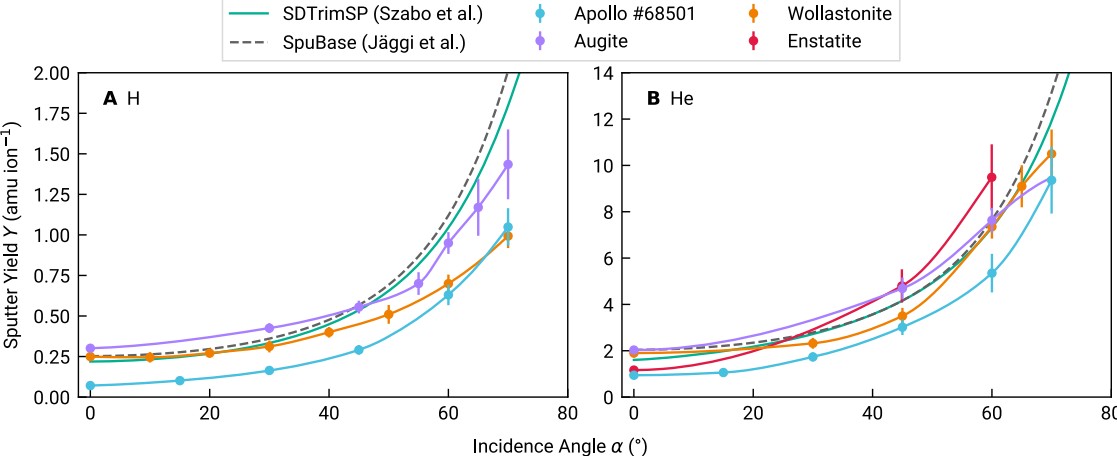

**Fig. 3 | Compilation of experimentally measured sputter yields Y over incidence angle α. A** Sputter yields under H ion bombardment. **B** Sputter yields for He ion bombardment. Data are shown for both this study (blue circles) and previous investigations using analogs (augite[74], purple circles; wollastonite[22,23], orange circles; and enstatite[24], red circles), compared to two approaches of SDTrimSP simulations[22,67] (grey dashed and teal solid lines, respectively) applied to the composition of the Apollo

16 sample 68501. For all cases, data are compared for flat samples. Connecting lines between experimental data are interpolated using a monotonic cubic spline to guide the eye. Experimental error bars are estimated from ion current fluctuations during the measurements and from the quality of fits to the QCM resonance frequency (this study), or taken from literature for the case of the previously published data.

## Table 1 | Sample compositions

|  | Thin film | Pellet | Literature |
|---|---|---|---|
| O | 61.0 ± 0.6 | 58.6 ± 0.6 | 61.0 |
| Si | 14.3 ± 0.4 | 17.0 ± 0.5 | 16.3 |
| Al | 9.96 ± 0.4 | 11.9 ± 0.5 | 11.3 |
| Ca | 8.14 ± 0.2 | 7.26 ± 0.2 | 5.93 |
| Mg | 3.27 ± 0.1 | 3.1 ± 0.1 | 3.37 |
| Fe | 2.76 ± 0.1 | 1.47 ± 0.1 | 1.65 |
| Ti | 0.45 ± 0.1 | 0.18 ± 0.1 | 0.16 |

Concentrations of the main components of the samples in at.% obtained by combining ToF-ERDA, RBS, and PIXE. For comparison, literature data are given[83,86].

correlation between this decrease and the mean of the surface inclination angle distribution[69]. Furthermore, this result was reinforced by an analytical investigation arriving at the same conclusion[70]. In this work, the pellet roughness (cf. Sample characterization) is comparable to the one in the mentioned enstatite study[24], as are the measured sputter yield reductions. In a further step, Cupak et al.[69] provide a Monte-Carlo-style algorithm called SPRAY that allows the calculation of sputter yields from atomic force microscopy (AFM) images of a given sample if the flat surface sputter yields are known. That way, any deviations from flat surface sputter yields are unambiguously attributable to surface morphology, as no other simulation parameter is varied. We found excellent agreement to our experimental results using this approach as well, pointing towards surface roughness as the main driver behind the observed sputter yield reduction for the pellet samples. Moreover, these additional simulation results also match with the SDTrimSP-3D data. We are thus confident that the 3D simulations capture the surface structure effects well, once the initial material-dependent over-estimation is corrected. We give the SPRAY results and a more in-depth description of the code in Supplementary Fig. 1 and the Supplementary Discussion. Additionally, the observed matching sputter yields for α = 45° and the reduction to about half of the thin film sputter yield at 60° match well with analytical predictions for the given roughness[70].

In contrast to these morphology effects, we do not expect crystallinity to play a role in the sputter yield modifications. While it is known that crystal structure has an effect on the sputtering properties of a sample[71] and that the sputtering behavior of amorphous and polycrystalline samples is not necessarily the same[72], these considerations are irrelevant in the context

of this study: Our flat samples are amorphous by the nature of their production process[23,28]. The pellets were pressed not from pristine minerals, but rather from regolith that naturally expresses amorphous rims around its crystalline sample fraction. Although fresh surfaces might have been created by breaking grains during the pellet pressing process, a 4 keV He fluence of $7.31 \times 10^{17}$ cm$^{-2}$ was applied to the samples during the first preparatory irradiation. It was shown that a 4 keV He fluence of $5 \times 10^{16}$ cm$^{-2}$ is sufficient to amorphize a rim of olivine ((Mg,Fe)$_2$SiO$_4$) with a resulting thickness of several 10 nm[73]. We therefore deem both the thin film and the pellet experimental results comparable to the (amorphous) BCA simulations both in the 1D and 3D configuration, and suitable to benchmark these very numerical results. Sputter yield data for the rough pellets match excellently between experiments and SDTrimSP-3D simulations after applying the same correction factor determined in Fig. 1 from flat sample data. It is thus justified to apply the same procedure to the porous regolith structures. Consequently, these regolith sputter yields should be used as realistic supply rates when modeling the lunar exosphere formation by solar wind ion impact. The fact that these sputter yields are notably lower than assumptions after models without experimental validation is, moreover, well in line with the findings that micrometeoroid impact vaporization might be the more dominant driver of particle ejection into the lunar exosphere[44].

In the past years, various studies reported sputter yields for lunar analog materials that were measured using the same method as described in this paper, where mass changes of flat, amorphous films were resolved by means of a QCM. Figure 3 compares the Apollo 68501 laboratory sputter yields from this work to data from wollastonite (CaSiO$_3$)[22,23], augite ((Ca,Mg, Fe)$_2$Si$_2$O$_6$)[74], and enstatite (MgSiO$_3$)[24]. In addition, the yields predicted by both SpuBase and SDTrimSP using the parameters by Szabo et al.[22] without scaling are shown. Both the regolith and the mineral samples share a similar composition with O and Si abundances of roughly 60 at.% and 20 at.%, respectively. As the sputter yield in the equilibrium is governed by the bulk stoichiometry, one would expect similar total mass yields across these samples, with the differences arriving from the variation of the metal species. This is the case, and most of the single-mineral data points lie between our Apollo data and the model predictions for the regolith composition. Moreover, this is also the reason why the total mass sputter yield is rather robust against slight deviations in sample composition: A variation in abundance of two or three percentage points of a given species will not manifest in resolvable changes of the mass yield. Even though some components in our samples, mostly minor in abundance, might deviate within

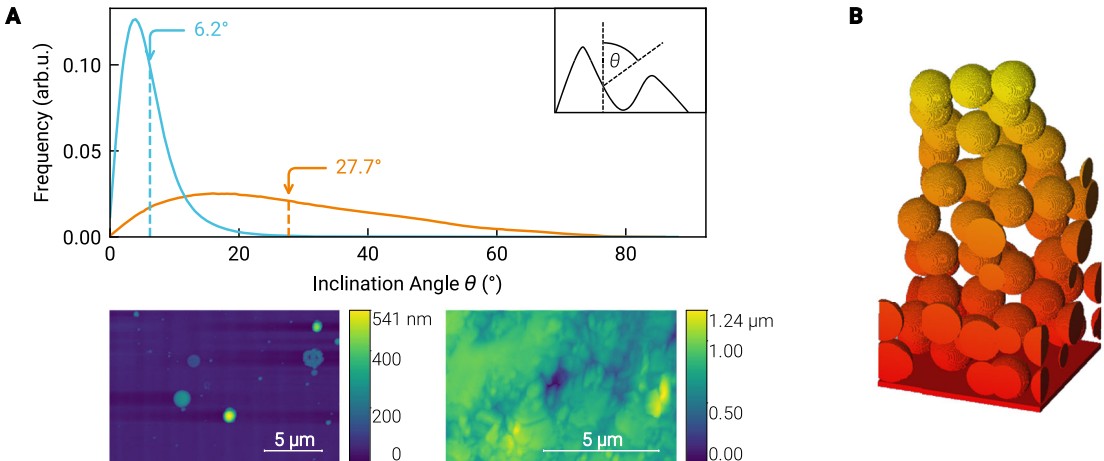

**Fig. 4 | Visualization of sample properties used during this study. A** AFM images of the thin film sample (left image in the bottom row) and the pressed pellet (right image in the bottom row). The top panel shows the surface inclination angle distribution for both experimentally used sample types, thin film (blue) and pellet (orange). The means of the distributions are marked by the respective vertical dashed lines and arrows. The inset defines the local inclination angle $\theta$ as the angle between the local and the global surface normal. **B** Example of the regolith structure with a porosity of about 0.8[40], which was used to simulate the sputter yield for the regolith case.

this range from the literature concentrations (cf. Table 1 and Section Sample characterization), this does not alter the measured sputter yields, because these species contribute little to the total yield. We further elaborate on this point in Supplementary Figs. 2 and 3.

Note that the yields measured from the lunar material are the lowest across all numerical and laboratory data sets (Fig. 3). While the models work well for individual minerals[46], they overestimate data for the case of the more complex Apollo soil. Sample roughness and composition cannot be the reason for this discrepancy, as these parameters are well-controlled experimentally across all these studies and comparable to the simulated cases. A more likely explanation could be the formation of bonds that exceed the ones typically found within an amorphous silicate layer. For example, both the compound binding model[46] and the Szabo et al.[22] approach reach the best agreement with laboratory data, assuming the importance of oxygen in the bond structure, either considering the oxide formation energy or by directly increasing the oxygen binding energy, followed by averaging of all the binding energies. This neglects bonds that could form between species of the different minerals. Should those bonds be stronger than the ones found in the bulk material, then the resulting higher binding energy would provide an explanation for the lower sputter yields. Another way of reducing yields would be by decreasing the target density and consequently increasing the binary collision mean free path. It is unclear why these effects are not found in glassy thin films of single mineral analogs. We propose that the high number of components in the Apollo sample would favor the formation of either longer (lower density) or stronger bonds (higher BEs) than are found in its components. This is mirrored in the way SpuBase handles such complex materials: Rather than assuming a glass of homogeneous composition (as is the case for the film on the QCMs and more standard SDTrimSP models[22]), they are decomposed from constituent minerals. While the difference between experiments and models underlines the necessity for experimental validation, particularly for complex samples, the proposed explanation can be tested by measurements of ejecta energy distributions of both the Apollo samples and the individual minerals. In addition to the arguments in the previous sections, this highlights once more the necessity for laboratory studies on sputtered ejecta energy distributions.

## Methods
### Experimental methods
Sputter yields were measured using a quartz crystal microbalance (QCM) technique in two configurations. In the first configuration, a flat, vitreous thin film deposited onto a quartz resonator is used as a sample. Using this

QCM, mass changes of the film due to ion bombardment are resolved in real time from changes in the quartz resonance frequency, allowing a direct calculation of the sputter yield. This setup and technique have a resolution[75,76] down to $1 \times 10^{-11}$ gs$^{-1}$ and have been successfully used in previous studies on analog materials[22,24,28,56]. In addition to this common configuration, we applied the catcher QCM method[24,77] where a second QCM is mounted within the experimental vacuum vessel, facing the irradiated sample. During ion irradiation, it measures the mass that is deposited onto its surface at the current position. This catcher QCM can be rotated with respect to the common center axis of the setup, subsequently varying the angle between the sample and catcher surface normals. This extension of the classic QCM setup allows the angular distribution of the sputtered ejecta to be probed. When applying this method to the above-described thin film QCMs as samples, both the sputter yields and the corresponding angular distributions of ejecta are measured simultaneously. Using these measurements as references, the catcher technique allows to reconstruct sputter yields of bulk samples of the same material, where otherwise no direct information would be available.

The ion beam irradiation setup has already been used in previous studies[23,24]. It consists of a 14.5 GHz electron cyclotron resonance ion source and an $m/q$ separation achieved via a magnetic sector field[78]. A set of computer-controlled deflection plates in front of the first aperture is used for switching the ion beam on/off electronically without moving parts, to minimize interference with the sensitive QCM signal. Scanning plates are used to ensure homogeneous sample irradiation. Furthermore, a Prevac FS40A1 electron flood source (up to 100 $\mu$A low energy electrons, <20 eV) was used to prevent charging of the insulating pellets due to the impinging ion beam. The 1 keV amu$^{-1}$ H data were obtained from double-energy $H_2$ irradiations. This is a common practice, and the underlying assumption is that a 2 keV hydrogen molecule is dissociated at the surface and acts as two independent hydrogen atoms of 1 keV each[23]. In the presented energy regime where sputtering is dominated by linear collision cascades, no effects are expected to arise from the molecular structure of the projectiles. Non-linearities occur at much lower energies[79], or considerably higher ones[80]. Moreover, this has been experimentally verified[28,81] and numerically checked by means of MD simulations[82]. Typically achieved ion fluxes are in the order of $3 \times 10^{12}$ cm$^{-2}$ s$^{-1}$ for He$^+$ and $1 \times 10^{13}$ cm$^{-2}$ s$^{-1}$ for $H_2^+$.

### Sample characterization
All samples used in this study were produced from Apollo soil 68051. This material is a mature (Is/FeO = 85) specimen collected during the Apollo 16

mission with an agglutinate content of about 38% and average grain sizes of $\approx 100\,\mu m$[83]. To study the influence of surface morphology, samples with rough surfaces were prepared in the form of pellets pressed from the lunar regolith. For this purpose, we used circular stainless-steel holders into which a layer of KBr was pressed to increase the cohesion between the sample and the back plate. Onto this interlayer, we then pressed the lunar sample. The pellet preparation process is well documented for a range of individual minerals[29], including the press specifications and illustrative photos of the pellets, both in press and after successful preparation. However, in contrast to these analog mineral pellets, we did not filter by grain size to keep the sample as representative as possible.

One such regolith pellet was subsequently used as a donor in pulsed laser deposition to grow flat thin films on QCM substrates. The depositions were performed under an $O_2$ atmosphere of $4 \times 10^{-2}$ mbar to achieve stoichiometric oxygen concentration in the resulting film. A KrF excimer laser was used with a wavelength of 248 nm and a pulse frequency of 5 Hz at a pulse energy of 400 mJ per pulse.

The chemical composition of both sample types was analyzed using a combination of ion beam analysis techniques[84]: time-of-flight elastic recoil detection analysis (ToF-ERDA), Rutherford backscattering spectrometry (RBS), and particle-induced X-ray emission (PIXE). ToF-ERDA was carried out using a combined anode gas ionization chamber and time-of-flight as detector with a primary beam of 36 MeV $^{127}$I$^{8+}$ and an incident angle of 67.5°. To unambiguously distinguish signals from species with similar atomic masses (i.e., Al and Si, K and Ca, Cr and Fe), RBS, simultaneously to PIXE was performed using He at 2 MeV and 5.5 MeV as primary beam at normal incidence. While RBS provides accurate quantification of Al and Si, PIXE is able to detect the presence of K and Cr in significantly smaller amounts compared to Ca and Fe, respectively. Finally, $\mu$-beam RBS/PIXE was used to verify the lateral homogeneity of the samples using 4 MeV He as a primary beam[85]. Different regions across the entire samples (including center and edge regions) were analyzed using a beam spot of $4\,\mu m$ to $5\,\mu m$ (1 mm × 1mm area per analysis). Results indicate that the composition of the pellet and thin film is homogeneously distributed along the samples. The atomic concentration of the main components observed in the samples is presented in Table 1 and agree mostly with literature[83,86], apart from some slightly off-stoichiometric concentrations for the heavier elements like Ca, Fe, or Ti. The influence of these deviations is discussed above as well as in the Supplementary Information in Supplementary Figs. 2 and 3.

In addition to the chemical analysis, AFM images were taken to characterize the surface roughness of both the thin film and pellet samples. No change in surface roughness was found after the ion beam experiments. The surface roughness was quantified using the surface inclination angle distribution method[69], the results of which are presented in Fig. 4. The bottom row in Fig. 4A shows AFM images of the thin film (left) and the pellet sample (right) and illustrates the difference in roughness: The film sample is generally flat as indicated by the uniformly colored surface base level with the exception of some particles that formed during the pulsed laser deposition. This difference in roughness is quantified by the respective surface inclination angle distribution and their means in the top panel, where the blue and orange lines denote the thin film and pellet, respectively.

Figure 4B shows a model of the porous regolith-structured sample used for the simulations. These structures were created using the parameters best fitting to reproduce ENA emission from backscattered solar wind protons[40,41]. The porosity is accordingly defined as the fraction of empty space between the regolith grains and the volume of the simulation cell from its lower boundary to the topmost grain.

## Computational modeling

Simulations of solar wind ion sputtering were carried out using SDTrimSP[31] (version 6.06) and SRIM[30] 2013. These codes employ the binary collision approximation that assumes collisions to involve only two particles at a given time, allowing for efficient calculation of energy and momentum transfer between collision partners[87]. Subsequently, the impactor and generated recoils are traced on their paths through the sample until their energies fall below a threshold. The resulting output includes information on the sputter yield as well as the ejecta angular distributions and energy distributions. Compared to SRIM, SDTrimSP allows for variations of more parameters and underlying physical models. Because the model assumes amorphous samples, these simulated data are directly comparable to the experimental results from the thin film QCM irradiations.

SRIM simulations were carried out using the damage calculation model "Detailed Calculation with full Damage Cascades". Apart from that, no other adaptations were made. For SDTrimSP with its default settings, simulations were run dynamically, accounting for fluence-dependent changes in surface composition. This is generally favored for compound targets under solar wind bombardment[34]. Other recommendations include the use of a mixed ion beam (96% H, 4% He) as well as using distributions for incidence angles and energies[34]. Due to the angle-resolved comparison to experimental data obtained from monatomic projectiles of a well-defined energy, we did not apply these. Finally, it is proposed to use adapted SBEs, if known[34]. Such are, however, not available for the studied sample to the best of our knowledge.

In the approach published by Szabo et al.[22], the surface binding model in SDTrimSP is set to isbv = 2, thereby averaging the SBEs of the constituent species. The oxygen SBE is increased to 6.5 eV, and the sample density is set to reflect the actual material density, in our case 3.1 g cm$^{-3}$ as per available literature[88].

In addition to the 1D simulations, we applied SDTrimSP-3D[89] (versions 1.21 and 1.22) to quantify how the surface morphology influences the sputter yield, as roughness and porosity are known to play an important role in the interaction of ions with materials[24,40,69,90]. We carried out the 3D calculations for the surfaces of the rough pellets as given by AFM images using the best fitting parameter set from the 1D cases. Furthermore, a porous regolith model was implemented[40]. These SDTrimSP-3D regolith simulations have already been shown to reproduce reflected neutral H spectra measured by Chandrayaan-1, strongly indicating an accurate description of the ion-regolith interaction in the model. Using this approach, the influences of roughness and porosity on the sputter yield compared to a flat surface can be untangled.

SDTrimSP and the adapted hybrid and compound binding energy model[46] also form the basis for SpuBase[67,68], a database of simulations already performed for flat surface samples. SpuBase offers sputter yields, ejecta angular and energy distributions for a wide variety of minerals. Results for bulk samples of a given atomic composition are then superposed from the constituent minerals. The improved hybrid binding model for compounds[46] (HB-C) is used in the calculations for the individual minerals, offering an improvement over arbitrary adaptation. However, data are only available for the 1D case, and the effects of surface morphology cannot be studied. The HB-C model also cannot be used directly in stand-alone SDTrimSP simulations, as the high number of components used in this study is not supported.

## Data availability

All data presented are openly accessible under https://doi.org/10.48436/zv721-mkb07.

## Code availability

SDTrimSP and its variants are available under an academic license or a commercial license from the Max Planck Institute for Plasma Physics (sdtrimsp@ipp.mpg.de) or Max-Planck-Innovation GmbH (info@max-planck-innovation.de), respectively. SpuBase is freely available[68].

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

## Acknowledgements

This research was funded in whole or in part by the Austrian Science Fund (FWF) [https://doi.org/10.55776/I4101]. For open access purposes, the author has applied a CC BY public copyright license to any author accepted manuscript version arising from this submission. Funding was also provided by the Swiss National Science Foundation Fund (200021L_182771/1, P500PT_217998). J.B. acknowledges financial support by KKKÖ of ÖAW. The authors gratefully acknowledge support from NASA's Solar System Exploration Research Virtual Institute (SSERVI) via the LEADER team, grant No. 80NSSC20M0060. We are very grateful to NASA for providing a lunar

regolith sample collected during the Apollo 16 mission via the Lunar Sample Request program (https://curator.jsc.nasa.gov/lunar/sampreq/requests.cfm). Ion beam analysis of the lunar material at UU was supported by the RADIATE project under the Grant Agreement 824096 from the EU Research and Innovation program HORIZON 2020. Accelerator operation at Uppsala University is supported by the Swedish Research Council VR-RFI (contract #2019_00191). The computational results presented have been achieved [in part] using the Vienna Scientific Cluster (VSC).

## Author contributions

J.B. and H.B. designed the study, carried out ion beam irradiation experiments, simulations, and sample analysis, and wrote the original draft. P.S.S. performed simulations, designed the 3D regolith model, and conceptualized the study. N.J. prepared samples, performed simulations, and conceptualized the study. L.F. carried out ion irradiation experiments. A.N. prepared samples and performed the pulsed laser depositions. M.F., G.N., E.P,. and D.P. performed the IBA analysis and evaluation of the sample composition. A.M. provided the SDTrimSP simulation suite. D.P., R.A.W., P.W., A.G., and F.A. supervised the study and acquired funding. In addition to these contributions, all authors reviewed and edited the original draft.

## Competing interests

The authors declare no competing interests.
