## [Transparent Peer Review file · Communications Earth & Environment]

Solar wind erosion of lunar regolith is suppressed by surface morphology and regolith properties

Corresponding Author: Mr Johannes Brötzner

Version 0:

Decision Letter:

Dear Mr Brötzner,

Please allow us to sincerely apologise for the long delay in sending a decision on your manuscript titled "Revealing Solar Wind Erosion of Lunar Regolith through High-Precision Experiments and 3D Modeling". It has now been seen by 2 reviewers, and we include their comments at the end of this message. They find your work of interest, but some important points are raised. We are interested in the possibility of publishing your study in Communications Earth & Environment, but would like to consider your responses to these concerns and assess a revised manuscript before we make a final decision on publication.

We therefore invite you to revise and resubmit your manuscript, along with a point-by-point response that takes into account the points raised. Please highlight all changes in the manuscript text file.

In particular, please ensure that in the revised manuscript you fully and clearly explain all aspects of your experimental approach and the characteristics of the samples you used.

Please submit your point-by-point responses as a separate file, distinct from your cover letter where you can add responses to the Editors' comments that you do not want to be made available to the reviewers. Word files are preferred. We recommend that any figures, tables or graphs that are included in the response to reviewers are also included in the main article or Supplementary Information.

Please use the following link to submit your revised manuscript, point-by-point response to the referees' comments (which should be in a separate document to any cover letter), a tracked-changes version of the manuscript (as a PDF file) and the completed checklist:

Link Redacted

We hope to receive your revised paper within six weeks; please let us know if you aren't able to submit it within this time so that we can discuss how best to proceed. If we don't hear from you, and the revision process takes significantly longer, we may close your file. In this event, we will still be happy to reconsider your paper at a later date, as long as nothing similar has been accepted for publication at Communications Earth & Environment or published elsewhere in the meantime.

Please do not hesitate to contact us if you have any questions or would like to discuss these revisions further. We look forward to seeing the revised manuscript and thank you for the opportunity to review your work.

Best regards,

Joe Aslin

Deputy Editor,
Communications Earth & Environment

Consulting Editor,
Communications Sustainability

<https://www.nature.com/commsenv/>
Twitter: @CommsEarth

EDITORIAL POLICIES AND FORMATTING

Editorial Policy: [Policy requirements](https://www.nature.com/documents/nr-editorial-policy-checklist.pdf) (Download the link to your computer as a PDF.)

- Behavioural and social science
- Ecological, evolutionary & environmental sciences
- Life sciences

<https://www.nature.com/documents/nr-reporting-summary.zip>

Furthermore, please align your manuscript with our format requirements, which are summarized on the following checklist: [Communications Earth & Environment formatting checklist](https://www.nature.com/documents/commsj-phys-style-formatting-checklist-article.pdf)

and also in our style and formatting guide [Communications Earth & Environment formatting guide](https://www.nature.com/documents/commsj-phys-style-formatting-guide-accept.pdf) .

***** DATA:** Communications Earth & Environment endorses the principles of the Enabling FAIR data project (<http://www.copdess.org/enabling-fair-data-project/>). We ask authors to make the data that support their conclusions available in permanent, publically accessible data repositories. (Please contact the editor if you are unable to make your data available).

All Communications Earth & Environment manuscripts must include a section titled "Data Availability" at the end of the Methods section or main text (if no Methods). More information on this policy, is available at <http://www.nature.com/authors/policies/data/data-availability-statements-data-citations.pdf>.

If a community resource is unavailable, data can be submitted to generalist repositories such as [figshare](https://figshare.com/) or [Dryad Digital Repository](http://datadryad.org/). Please provide a unique identifier for the data (for example a DOI or a permanent URL) in the data availability statement, if possible. If the repository does not provide identifiers, we encourage authors to supply the search terms that will return the data. For data that have been obtained from publically available sources, please provide a URL and the specific data product name in the data availability statement. Data with a DOI should be further cited in the methods reference section.

REVIEWER COMMENTS:

Reviewer #1 (Remarks to the Author):

This manuscript reports a detailed analysis of sputtering yields simulated through different codes compared to the yields obtained experimentally from a lunar flat and in pellet samples. The results highlight the difference between the different codes and between different surface roughness. The importance of considering more representative mineralogical mixtures from real lunar samples instead of samples obtained from one single mineralogy obtained from Earth analogues is also an interesting conclusion that opens up new experimental directions. The manuscript is clear and well written. I have minor comments that could improve some parts.

Minor comments:

Table I and discussion: contrary on what is stated in lines 317-318 (The atomic concentration of the main components observed in the samples are presented in Table I and agree well with literature [84, 85]), the compositional values are not fully in agreement; in fact, (for example, Fe → film 2.76 ± 0.1 , pellet 1.47 ± 0.1 , lit. 1.65) these values are not in agreement within the uncertainty. Is there any criteria on the value within which the difference could be considered negligible?

in figure 3, the used colors in the plots are difficult to distinguish. Light blue is used for different cases, the pink and violet are quite similar, making difficult to discriminate the right curve. The interpolated curves are not just linear interpolations. Are they a polynomial interpolation?

Reviewer #2 (Remarks to the Author):

The paper reports experimental measurements and modeling estimates of the lunar regolith's sputtering yield under solar wind bombardment, a crucial parameter for quantitatively estimating the weathering rate of the lunar surface, the relative importance of sputtering versus micrometeoritic bombardment to the weathering, the escape rate of energetic (i.e. sputtered) atoms from the surface to space, and the source of material to the lunar exosphere. The study uses Apollo regolith samples to measure/model the yield versus ion incidence angle and morphology (flat vs rough), for 1 keV/amu H and He ions. The experimental measurements give yields substantially lower, by almost an order of magnitude, than model based estimates that do not take into consideration material composition and regolith morphology. The paper shows that more advanced modeling approaches that include these effects produce lower sputtering yields in better agreement with experiment than basic SRIM simulations. However, the updated models still disagree substantially enough with experiment to require additional scaling factors to match the data. The findings suggest that sputter-produced erosion/loss of the surface material, and the source of material to the lunar exosphere, may be substantially below previous estimates. Overall I think the findings are significant, but to assess their actual impact I have a few questions (below, in rough order of importance), which may merit clarification in the manuscript.

I am trying to understand what the paper is (or is not) intending to recommend regarding how to apply these data. Presumably the next step might be to use the SDTrimSP model to other energies, so that the full solar wind flux / energy spectrum can be integrated to estimate the Moon's total sputtered flux. To this point, is it justifiable to apply the 0.47 and 0.66 scaling factors to energies other than 1 keV/amu ?

As a general comment, some sort of sentence, maybe at the end of (or beginning of) the introduction that explained exactly (I mean specifically) what was done that is new, would have been extremely helpful to me as the reader. For example, something like: "In this paper, we report new laboratory sputtering yield measurements from 1 keV/amu H and He irradiated flat and rough pellet lunar material samples extracted from Apollo 16 sample 68501, and compare against the results of new SRIM, SpuBase, and SDTrimSP models."

Maybe I missed something, but I am confused about the red line in Figure 2. Was an unprocessed Apollo sample (porous regolith) also measured in the lab ? The red line appears as though it's connecting points but no points are shown. Section IV.A. only talks about experiments on thin films and pellets, so it was not clear to me what was done with the porous regolith (is this just a model calculation?).

It's hard for me to assess the paper's claim that morphology is an important factor for the yield, since Fig. 2 seems to show only a minor difference between the flat and rough pellet cases, and (again) it's unclear to me what the regolith case is supposed to be in Fig 2 (my previous question).

It's not explained what a "pellet" actually is, how it's prepared, etc. I understand that it's explained in reference 29, but considering that pellets are one of the main cases studied, does it not merit at least a sentence or two of explanation in the paper?

Figures 1 and 2: How are we to interpret the y-axes? Why are there two scales?

The thin film is described as flat (line 262), but why does the AFM image in Fig. 4 not appear flat? Presumably I am misinterpreting it, but it appears to me to have nearly zero thickness (according to the color scale) with a few discrete particles.

Figure 3. Should it not be clearly explained in the caption that this is showing the flat surface case?

Section IV.A. It's explained that a QCM is used in the thin film case to measure the sputtering yield. Is this also done for the rough pellet case? Is it not more difficult to get the QCM to resonate with crushed material (as compared to a PLD deposited film)? Or is only the catcher QCM being used for the pellet case? The paper was not very clear on this.

Minor edits: Line 189: "allowing to calculate" should be "allowing the calculation of". Also line 273: "allows to probe" should be something like "allows the angular distribution of the sputtered ejecta to be probed".

Communications Earth & Environment is committed to improving transparency in authorship. As part of our efforts in this direction, we are now requesting that all authors identified as 'corresponding author' create and link their Open Researcher and Contributor Identifier (ORCID) with their account on the Manuscript Tracking System prior to acceptance. ORCID helps the scientific community achieve unambiguous attribution of all scholarly contributions. You can create and link your ORCID from the home page of the Manuscript Tracking System by clicking on 'Modify my Springer Nature account' and following the instructions in the link below. Please also inform all co-authors that they can add their ORCID to their accounts and that they must do so prior to acceptance.

Version 1:

Decision Letter:

Dear Mr Brötzner,

Your manuscript titled "Revealing Solar Wind Erosion of Lunar Regolith through High-Precision Experiments and 3D Modeling" has now been seen by our reviewers, whose comments appear below. In light of their advice we are delighted to say that we are happy, in principle, to publish a suitably revised version in Communications Earth & Environment.

We therefore invite you to edit your manuscript to comply with our format requirements and to maximise the accessibility and therefore the impact of your work.

EDITORIAL REQUESTS:

*****Please take care to match our formatting and policy requirements. We will check revised manuscript and return manuscripts that do not comply. Such requests will lead to delays. *****

SUBMISSION INFORMATION:

OPEN ACCESS:

Communications Earth & Environment is a fully open access journal. Articles are made freely accessible on publication. For

further information about article processing charges, open access funding, and advice and support from Nature Research, please visit <https://www.nature.com/commsenv/open-access>

Link Redacted

Best regards,

Joe Aslin

Deputy Editor,
Communications Earth & Environment

Consulting Editor,
Communications Sustainability

<https://www.nature.com/commsenv/>

Twitter: @CommsEarth

REVIEWERS' COMMENTS:

Reviewer #1 (Remarks to the Author):

The authors provided exhaustive explanations to my questions.
Also the clarifications after the other referee's concerns, made the manuscript clearer and ready for publication.

Reviewer #2 (Remarks to the Author):

The expectations that phenomena such as preferential sputtering may alter surface composition leading to reduced sputtering yields, and that rough morphology may produce flat yields versus angle, are not new, but it is helpful for understanding the lunar exosphere to have evidence for these effects on the moon with more advanced models and especially experiments as done here. Future experiments on regoliths and at other energies would I think bolster the work further. In regards to this paper, all of my questions have been addressed.

Response to reviewers

We are grateful to the reviewers for their constructive and encouraging feedback, which we believe has helped to further improve the quality and clarity of our work.

Reviewer #1

(Remarks to the Author):

This manuscript reports a detailed analysis of sputtering yields simulated through different codes compared to the yields obtained experimentally from a lunar flat and in pellet samples. The results highlight the difference between the different codes and between different surface roughness. The importance of considering more representative mineralogical mixtures from real lunar samples instead of samples obtained from one single mineralogy obtained from Earth analogues is also an interesting conclusion that opens up new experimental directions. The manuscript is clear and well written. I have minor comments that could improve some parts.

We thank the reviewer for the constructive and encouraging feedback. Please see our responses to the remaining comments below.

Minor comments:

Table I and discussion: contrary on what is stated in lines 317-318 (The atomic concentration of the main components observed in the samples are presented in Table I and agree well with literature [84, 85]), the compositional values are not fully in agreement; in fact, (for example, Fe -> film 2.76 \pm 0.1 , pellet 1.47 \pm 0.1 , lit. 1.65) these values are not in agreement within the uncertainty. Is there any criteria on the value within which the difference could be considered negligible?

This is indeed a good point that we admittedly rushed over too quickly in our initial submission. The reason that this discrepancy does not take away from our results lies in the behaviour of the steady state sputter yield: In the equilibrium, the sputtered particle fluxes behave according to bulk stoichiometry, while the surface composition has been altered due to preferential sputtering and a depletion of the volatile species. We included an additional reference for this in the manuscript and adapted the wording in the Results Section, in line with the question about the two y-axes and their units by reviewer #2. As Fe or Ti are not very abundant in our samples, only a few at.%, their contribution to the ejecta flux is also low. Even though they are the heaviest components in our samples, their mass is not sufficient to significantly alter the total mass sputtered per ion, which is the physical quantity that we measure.

To check this statement, we carried out new simulations: First, according to the literature composition. However, we also focussed on the Fe the reviewer pointed out in particular. We decreased the iron content by 1.1 at.% points to make it match to the literature value. As SDTrimSP requires the composition to sum to unity, we studied 3 cases:

1. We took the 1.1 at.% and spread them evenly across the remaining components.
2. We added the 1.1 at.% to the abundance of O, the element with the highest concentration.
3. We added the 1.1 at.% entirely to Ca, the heaviest element (with an abundance >0.5 at.%) where one would thus expect the most significant influence on the total sputter yield.

The SDTrimSP simulation results for all cases are pretty much identical and we could not experimentally resolve any difference between them. So even though the deviations from the literature composition could mineralogically describe a slightly different sample, the influence on the sputter yield is negligible. We include this figure along with a brief discussion in the newly created supplementary materials.

On a similar note, SpuBase comes with example compositions for both the lunar highland anorthosite and the mare basalts. These differ in their Ti and Fe content by roughly 3 at.% points and 0.6 at.% points, respectively. Nonetheless, the variation in total sputter yield is still significantly smaller than our experimental uncertainty. See the figure above, which we also include in the supplementary materials. A criterion for negligibility of these discrepancies could thus be in the context of our QCM measurements that the composition variation has to concern elements that are either low enough in abundance or low enough in atomic mass such that they do not contribute significantly to the total removed mass per ion. Exchanging abundant *and* heavy species leads then to the larger differences that are shown for different minerals in Fig. 3 of the manuscript.

In the manuscript, we adapted the Sample Characterization:

The atomic concentration of the main components observed in the samples are presented in Table 1 and agree well mostly with literature [85,86], apart from some slightly off-stoichiometric concentrations

for the heavier elements like Ca, Fe or Ti. The influence of these deviations is discussed in Section III as well as the Supplementary Materials in Supplementary Figs. 2 and 3.

We also added to the Discussion:

Moreover, this is also the reason why the total mass sputter yield is rather robust against slight deviations in sample composition: A variation in abundance of two or three percentage points of a given species will not manifest in resolvable changes of the mass yield. Even though some components in our samples, mostly minor in abundance, might deviate within this range from the literature concentrations (cf. Table 1 in Section IVB), this does not alter the measured sputter yields, because these species contribute little to the total yield. We further elaborate in this point in Supplementary Figs. 2 and 3.

Because this notion is now more thoroughly discussed, we omitted the following sentence from the next paragraph in the Discussion:

~~Moreover, we also varied the composition input to the simulations within the error bars from the sample analysis (Table 1 in Section IVB) and found only an insignificant level of deviation in the results.~~

in figure 3, the used colors in the plots are difficult to distinguish. Light blue is used for different cases, the pink and violet are quite similar, making difficult to discriminate the right curve. The interpolated curves are not just linear interpolations. Are they a polynomial interpolation?

Thank you. We wanted to keep the lines styled as they appeared in previous figures, but we realize that this resulted in a loss of clarity. We therefore changed the color of the SpuBase line to a dark grey and the SDTrimSP line to a teal color. This should make the curves better distinguishable.

We used a cubic spline to interpolate the data points. We have included this information in the figure caption, such that it now contains the following:

Connecting lines between experimental data are interpolated **using a monotonic cubic spline** to guide the eye.

Reviewer #2

(Remarks to the Author):

The paper reports experimental measurements and modeling estimates of the lunar regolith's sputtering yield under solar wind bombardment, a crucial parameter for quantitatively estimating the weathering rate of the lunar surface, the relative importance of sputtering versus micrometeoritic bombardment to the weathering, the escape rate of energetic (i.e. sputtered) atoms from the surface to space, and the source of material to the lunar exosphere. The study uses Apollo regolith samples to measure/model the yield versus ion incidence angle and morphology (flat vs rough), for 1 keV/amu H and He ions. The experimental measurements give yields substantially lower, by almost an order of magnitude, than model based estimates that do not take into consideration material composition and regolith morphology. The paper shows that more advanced modeling approaches that include these effects produce lower sputtering yields in better agreement with experiment than basic SRIM simulations. However, the updated models still disagree substantially enough with experiment to require additional scaling factors to match the data. The findings suggest that sputter-produced erosion/loss of the surface material, and the source of material to the lunar exosphere, may be substantially below previous estimates. Overall I think the findings are significant, but to assess their actual impact I have a few questions (below, in rough order of importance), which may merit clarification in the manuscript.

We thank the reviewer for their generally positive assessment of our work and do our best to address all remaining questions below. Upon reading the reviewer's comments, we realized that many questions concern a lack of clarity. We added clarifications/elaborations into the manuscript where appropriate and requested.

I am trying to understand what the paper is (or is not) intending to recommend regarding how to apply these data. Presumably the next step might be to use the SDTrimSP model to other energies, so that the full solar wind flux / energy spectrum can be integrated to estimate the Moon's total sputtered flux. To this point, is it justifiable to apply the 0.47 and 0.66 scaling factors to energies other than 1 keV/amu?

From a purely fundamental point of view, we want to convey that getting realistic sputter yields for exosphere modelling purposes requires consideration of realistic surface inputs and ideally experimental benchmarks for the material of interest. From an applied perspective, we provide numbers for both the hydrogen and helium that are applicable for a range of lunar latitudes and robust throughout the expected lunar geology (for this, see also our response to reviewer #1). To emphasise this point more, we elaborated on the closing paragraph of the Introduction:

Our findings provide realistic sputter yield estimates for actual lunar regolith of $7.3e-3$ atoms/ion and $7.6e-2$ atoms/ion for H and He, respectively, which are up to an order of magnitude smaller than previous estimates. ~~in line with the conclusions of [44] that micro-meteoroid impacts play a more significant role in lunar space weathering than previously recognized.~~ These values are largely independent from the ion incidence angle, i.e. the solar zenith angle, and thus valid over a wide range of lunar latitudes. They are furthermore robust against slight variations in sample composition and therefore representative for lunar geology beyond the specific Apollo sample 68501.

Regarding the scaling factors and the energy dependence: The necessary correction depends primarily on the sample material and the projectile. For instance, the model adaptations proposed in [22] are

shown therein to work for CaSiO_3 (for which they were derived) under 2 keV Ar bombardment as well as for 4 keV ^4He and 3 keV ^3He . Only for protons, further corrections are necessary due to their implantation and chemical sputtering [22]. We would therefore expect our scaling factors to be similarly applicable for a range of energies. On the other hand, this might not even be necessary; Morrissey et al. showed in [34] that when simulating sputtering by ions with energies distributed according to the slow solar wind energy distribution, the resulting yields are only up to 5% lower than for the approximation of 1 keV/amu. The energy is more important for proper descriptions of implantation and depth profiles of ion-induced damages. Of course, experiments to extend our findings to other energies are possible; We cautiously doubt, however, that they would bring novel benefits.

As a general comment, some sort of sentence, maybe at the end of (or beginning of) the introduction that explained exactly (I mean specifically) what was done that is new, would have been extremely helpful to me as the reader. For example, something like: "In this paper, we report new laboratory sputtering yield measurements from 1keV/amu H and He irradiated flat and rough pellet lunar material samples extracted from Apollo 16 sample 68501, and compare against the results of new SRIM, SpuBase, and SDTrimSP models."

We adapted the last paragraph of the introduction outlining the contents of the manuscript according to the suggestion. We also included clarifications about the porous model, in line with following comments. The last newly added sentence will further help clarify points from upcoming remarks.

(...) Our study combines experimental and numerical approaches to quantify sputter yields from actual lunar material (~~Apollo 16 sample 68501~~) rather than typically used analog minerals, irradiated with H and He ions **at solar wind velocities of ≈ 440 km/s. 1 keV/amu (corresponding to a velocity of ≈ 440 km/s).** **In particular, we report new laboratory sputter yield measurements for both flat and rough samples prepared from Apollo soil 68501 and irradiated by 1 keV/amu H and He, and compare against the results of SRIM, SpuBase, and SDTrimSP models.** **Moreover, we model the effect of regolith porosity in addition to rough, but compact, surface morphologies – an effect that is currently not accessible experimentally.** We demonstrate that the commonly used simulation codes (...)

Maybe I missed something, but I am confused about the red line in Figure 2. Was an unprocessed Apollo sample (porous regolith) also measured in the lab? The red line appears as though it's connecting points but no points are shown. Section IV.A. only talks about experiments on thin films and pellets, so it was not clear to me what was done with the porous regolith (is this just a model calculation?).

For the regolith case (the red line in Fig. 2), there are unfortunately no experimental data available. In our lab, the ion beam is extracted from the ECR source horizontally, and the samples are mounted on a vertically hanging manipulator. For obvious reasons, this is not possible for loose regolith without some form of handling, compacting and thus altering its properties.

We are therefore constrained to simulations only for this case. However, we are confident that even with the absence of experimental data in this case, the red lines describe physically valid and relevant sputter yields. On the one hand, the actual geometric model in use was already verified in previous studies by comparison to in-situ data from Chandrayaan-1 (references [40, 41] in the manuscript). On the other hand, at this point any material-dependent overestimation of sputter yields by SDTrimSP is already factored out through the application of the correction factors illustrated in Fig. 1. These factors were derived for the 1D case and are not changed anymore once roughness and porosity are introduced. In

that regard, the pellet study forms an important validation case: For the pellet, we have both measured and simulated sputter yields. After the application of the correction factors from the 1D simulations, the experimental and numerical data agree excellently. We can thus conclude that the 3D simulations can accurately model the influence of surface morphology on the sputter yield, once material-dependent shortcomings are addressed.

To address this, we adapted the paragraph concerning sample roughness in the Discussion Section, also partially in response to the remark about surface morphology:

(...) In a further step, Cupak et al. provide a Monte-Carlo-style algorithm called SPRAY that allows allowing the calculation of sputter yields from atomic force microscopy (AFM) images (...). We found excellent agreement to our experimental results using this approach as well, pointing towards surface roughness as the main driver behind the observed sputter yield reduction for the pellet samples. Moreover, these additional simulation results also match with the SDTrimSP-3D data. We are thus confident that the 3D simulations capture the surface structure effects well, once the initial material-dependent overestimation is corrected. We give the SPRAY results and a more in-depth description of the code in Supplementary Fig. 1 and the Supplementary Discussion. We are thus confident that the 3D simulations capture the surface structure effects well, once the initial material-dependent overestimation is corrected.

As SDTrimSP calculates sputter yields always for a given, but fixed, incidence angle, the regolith data are available for a discrete set of incidence angles (0° , 15° , 30° , 45° , 60° , 75° , 80° , 85°). As the porous 3D calculations are computationally more expensive than the much simpler 1D ones, the distance between angles is larger and only decreased for the grazing incidence regime. This leads to the look the reviewer mentioned. We decided to present the data with connecting lines to keep the same convention throughout the manuscript (lines for simulations, markers for experimental data).

To make the numerical nature of the regolith data clearer, we edited the corresponding paragraph in the Results section:

As sputter yield measurements for loose regolith powder are not feasible in our setup, we introduced porosity in SDTrimSP-3D calculations (red lines). In this case, ~~When porosity is introduced through the implementation of regolith structures in SDTrimSP-3D (red lines),~~ the same effect is observed (...)

It's hard for me to assess the paper's claim that morphology is an important factor for the yield, since Fig. 2 seems to show only a minor difference between the flat and rough pellet cases, and (again) it's unclear to me what the regolith case is supposed to be in Fig 2 (my previous question).

The concept that the surface structure influences the sputtering behaviour is not new and has been discussed for decades. See, for example, the following work by Küstner: M. Küstner et al., Nucl. Instrum. Methods Phys. Res. B 145 (1998) 320–331, doi:10.1016/S0168-583X(98)00399-1. While the data presented in our manuscript might not look too different between the sets from the flat and the rough samples, it is important to point out that we could only study one particular roughness, as determined by the pellet pressing process. A systematic study covering various degrees of roughness (expressed both in terms of the RMS roughness as well as the surface inclination distribution) is given by Cupak et al. in reference [69] of our manuscript. A supplementary analytical description can be found in [70]. From a fundamental physics perspective, the difference between the flat and rough sample sputter yields is

indeed not “minor”: Note that for both projectiles in Fig. 2, there is about a factor of 2 difference between the sample configurations under grazing incidence without overlapping uncertainties. For technical or commercial applications of sputtering, such as in preparing atomistically clean surfaces or sputter coating of materials, such a factor is very noticeable rather than negligible.

As outlined also in our response to the previous question, the yield reduction for the regolith case also comes from the surface structure of the sample, and we are confident that the simulation accurately models the effect of surface morphology. To further untangle the morphology influence from the BCA simulations, we employed the SPRAY code provided by and described in ref. [69]. We included this argument in the Discussion Section as well, but reiterate the main points here to clear up confusions: The SPRAY codes maps 1D simulations onto real surfaces (AFM images of the pellets in our case) and does not calculate the whole collision cascade again. If the 1D input is accurate (cf. Fig. 1 in the manuscript), any sputter yield deviations have to be from surface effects. To illustrate this point more comprehensibly, we added a figure and a description of the SPRAY code to the newly created supplementary materials. We also added a reference to the supplementary materials in the Discussion Section about the roughness influence.

We believe that the origin and meaning of the regolith line is now evident from our description in the last comment (above).

It's not explained what a "pellet" actually is, how it's prepared, etc. I understand that it's explained in reference 29, but considering that pellets are one of the main cases studied, does it not merit at least a sentence or two of explanation in the paper?

We changed the beginning of Section IIB:

A comparison of the total sputter yields for flat PLD films, rough pellets, and porous regolith samples is given in Fig. 2A and Fig. 2B for 1 keV/amu H and He, respectively. Just like in the previous subsection, experimental data for flat samples were obtained from PLD films. For measurements on rough surfaces, we pressed some of the original regolith sample in order to form stable pellets that can withstand mounting in the vacuum vessel. Experimental data for these were obtained by means of a catcher QCM (cf. Section IVA). As this is not possible for the porous regolith case, only simulated data are available for this instance. For preparation and characterization, the reader is referred to Section IVB. Note that for (...)

We also elaborated on the pellet production process in Section IVB:

All samples used in this study were produced from Apollo soil 68051. This material is a mature (Is/FeO = 85) specimen collected during the Apollo 16 mission with an agglutinate content of about 38% and average grain sizes of $\approx 100 \mu\text{m}$ [85]. To study the influence of surface morphology, samples with rough surfaces were prepared in the form of pellets pressed from the lunar regolith. For this purpose, we used circular stainless-steel holders into which a layer of KBr was pressed to increase the cohesion between the sample and the back plate. Onto this interlayer we then pressed the lunar sample. The pellet preparation process is described in greater detail in [29], including the press specifications and illustrative photos of the pellets, both in press and after successful preparation. However, in contrast to the analogue mineral pellets described therein, we did not filter by grain size to keep the sample as representative as possible. Pellets were pressed following the procedure in [29], but without filtering for grain sizes.

A layer of KBr was first pressed into the holder to increase cohesion between the regolith and the back-plate.

One such regolith pellet was subsequently used as a donor in pulsed laser deposition to grow flat thin films on QCM substrates. Thin films were deposited onto QCMs by means of pulsed laser deposition. This was done The depositions were performed under (...)

Figures 1 and 2: How are we to interpret the y-axes? Why are there two scales?

Both y-axes give sputter yields. In both figures and all subpanels therein, the left axis gives the sputter yield in atomic mass units per incident ion. This is the physical quantity that is directly accessible by the quartz crystal microbalance technique and therefore the most accurate representation of our measurements. On the other hand, BCA simulations typically give results on an atomistic basis, i.e. sputter yields will be given in ejected atoms per ion. This is usually the preferred unit also for exosphere modellers, which is why we opted to give both units.

Calculating the yields in amu/ion from atoms/ion is rather straightforward and follows from multiplication with the species' atomic mass and addition of all partial, species-specific sputter yields. The other direction is a little bit more involved; Due to the preferential sputtering of more volatile elements in compound materials, sputtering will dynamically change the surface composition until a steady state is reached. In this steady state, the surface composition will be different compared to the bulk composition, but the sputtered particle fluxes will follow the bulk stoichiometry. Given the total sputter yield and the bulk composition, one can thus also calculate a sputter yield in atoms/ion from our QCM measurements. The necessity of a fixed composition in this calculation is also the reason why in Figure 3, where different minerals are compared, we refrained from including a second y-axis. We elaborated on this point in the manuscript and added reference [45]. The corresponding paragraph in the Results Section now reads:

The sputter yields are given as function of incidence angle α in atomic mass units per incident ion (left axis of both panels) and atoms per incident ion under the assumption of stoichiometric particle fluxes (right axis of both panels). While our QCM technique (see Section IV A) directly measures the sputter yields as mass changes (left axes), simulations typically give them in atoms per ion. Sputtered atoms per ion is also usually the unit of choice for exosphere models (see, e.g., [12]), which is why we give both units wherever possible. Conversion of the experimental data to atoms per ion was carried out under the assumption that in the steady state, the elemental sputtered particle fluxes correspond to the bulk stoichiometry [22, 45].

We do realise, however, that the left label in Figure 1 mistakenly reads "Sputter Yield Y (atoms/ion)" where the unit should have been amu/ion. We corrected the axis label and thank the reviewer for their keen observation.

The thin film is described as flat (line 262), but why does the AFM image in Fig. 4 not appear flat? Presumably I am misinterpreting it, but it appears to me to have nearly zero thickness (according to the color scale) with a few discrete particles.

The thin film is indeed flat with some discrete particles that are a product of the pulsed laser deposition process. The 0-level of the colour code was chosen to represent the point of lowest elevation in the AFM image, i.e., the surface level of the film. Rather than a film of near-zero thickness, the image should thus be interpreted as dense, flat film with near-zero elevations from its surface; the exception to this being

the few particles. From the AFM images we also extracted the distribution of surface inclination angles with respect to the surface normal. For a perfectly flat surface, this distribution would have the shape of a delta-distribution centred at 0° . Our flat films have a mean located at 6.2° (as indicated in the figure), making them “flatter” by this metric compared to films in the conceptionally similar studies of Cupak et al. and Biber et al. (references [69] and [24] in the original manuscript, respectively).

We also know that the films are finitely thick and dense, coating the entire sensitive quartz area, rather than having nearly zero thickness. On the one hand, we have experience in our lab with dealing with such mineral films and their characterisation (see, i.e., references [22, 24, 28, 56] in the manuscript, but most importantly [23]). On the other hand, a partially exposed gold electrode of the QCM (the substrate of the films) would exhibit a much higher sputter yield due to high atomic mass of Au. However, we only see sputter yields increasing after applying sufficient fluence to erode the tens of nm of sample film.

To make things clearer in the paper, we added to the following sentence:

The film sample is generally flat **as indicated by the uniformly colored surface base level** with the exception of some particles that formed during the PLD deposition.

Figure 3. Should it not be clearly explained in the caption that this is showing the flat surface case?

We agree and include the following sentence with the figure caption:

For all cases, data are compared for flat samples.

Section IV.A. It's explained that a QCM is used in the thin film case to measure the sputtering yield. Is this also done for the rough pellet case? Is it not more difficult to get the QCM to resonate with crushed material (as compared to a PLD deposited film)? Or is only the catcher QCM being used for the pellet case? The paper was not very clear on this.

For the rough pellet case, the catcher QCM technique was used as an extension to the direct one. With the catcher configuration, we probe the angular distribution of the sputtered ejecta as a slice through the emission plume in the polar angle direction. However, this information alone is not sufficient to quantitatively reconstruct the sputter yields of a rough target: We experienced in the past that for compound targets, not every species necessarily has the same sticking probability. Therefore, integration over all polar angles of the measured distribution does not uniquely identify the rough sample sputter yields. We deal with this problem by alternating irradiations of rough samples and the thin films on QCM targets. For the latter, the sputter yields are directly measured, simultaneously to the angular distribution. A known sputter yield thus gives a known angular distribution. Therefore, the rough sample sputter yield is then determined by the flat sample sputter yield, scaled with the ratio of the integrated angular distributions of both sample types. This more involved experiment is the also the reason why measuring sputter yields of rough samples takes a lot longer, and consequently, we present fewer data points for the pellet case in Fig. 2.

More concisely, no crushed material was used directly on a QCM and the catcher QCM method had to be employed. Our research group started to develop this technique as early as 2017 (cf. [77] in the paper) with gold targets as a proof of principle and applied it to rough mono-elemental W targets [69] and finally minerals [24]. It is thus well-documented and reliable. An additional approach by another group with catcher QCMs covering also the azimuthal direction and a discussion about sticking

coefficients can be found in the following publication: C. Bu et al., J. Appl. Phys. 135 (2024) 035302, doi:10.1063/5.0184417.

To make it clearer that for the pellets, we used the catcher technique, we added a sentence to first appearance of these data in the Results section (see also our response to the pellet question above).

Minor edits: Line 189: “allowing to calculate” should be “allowing the calculation of”. Also line 273: “allows to probe” should be something like “allows the angular distribution of the sputtered ejecta to be probed”.

We rephrased these parts according to the reviewer’s suggestions.

Response to reviewers

We are grateful to the reviewers for their constructive and encouraging feedback. We provide our response to their final remarks below on this page. We keep their comments and our responses from the first round of review on the following pages.

Reviewer #1

The authors provided exhaustive explanations to my questions. Also the clarifications after the other referee's concerns, made the manuscript clearer and ready for publication.

Thank you for the positive assessment! We are glad to have clarified all initially raised matters.

Reviewer #2

The expectations that phenomena such as preferential sputtering may alter surface composition leading to reduced sputtering yields, and that rough morphology may produce flat yields versus angle, are not new, but it is helpful for understanding the lunar exosphere to have evidence for these effects on the moon with more advanced models and especially experiments as done here. Future experiments on regoliths and at other energies would I think bolster the work further. In regards to this paper, all of my questions have been addressed.

Thank you for recognizing the value of our modeling and experimental efforts. While preferential sputtering is indeed not a new concept, the extent to which altered surface chemistry in complex samples and porosity, rather than just roughness, affect sputter yields required direct investigation and comparison to more simple analogue samples. We appreciate the suggestion for future studies and are glad that all questions have been answered.

Reviewer #1

(Remarks to the Author):

This manuscript reports a detailed analysis of sputtering yields simulated through different codes compared to the yields obtained experimentally from a lunar flat and in pellet samples. The results highlight the difference between the different codes and between different surface roughness. The importance of considering more representative mineralogical mixtures from real lunar samples instead of samples obtained from one single mineralogy obtained from Earth analogues is also an interesting conclusion that opens up new experimental directions. The manuscript is clear and well written. I have minor comments that could improve some parts.

We thank the reviewer for the constructive and encouraging feedback. Please see our responses to the remaining comments below.

Minor comments:

Table I and discussion: contrary on what is stated in lines 317-318 (The atomic concentration of the main components observed in the samples are presented in Table I and agree well with literature [84, 85]), the compositional values are not fully in agreement; in fact, (for example, Fe -> film 2.76 +/-0.1 , pellet 1.47 +/-0.1 , lit. 1.65) these values are not in agreement within the uncertainty. Is there any criteria on the value within which the difference could be considered negligible?

This is indeed a good point that we admittedly rushed over too quickly in our initial submission. The reason that this discrepancy does not take away from our results lies in the behaviour of the steady state sputter yield: In the equilibrium, the sputtered particle fluxes behave according to bulk stoichiometry, while the surface composition has been altered due to preferential sputtering and a depletion of the volatile species. We included an additional reference for this in the manuscript and adapted the wording in the Results Section, in line with the question about the two y-axes and their units by reviewer #2. As Fe or Ti are not very abundant in our samples, only a few at.%, their contribution to the ejecta flux is also low. Even though they are the heaviest components in our samples, their mass is not sufficient to significantly alter the total mass sputtered per ion, which is the physical quantity that we measure.

To check this statement, we carried out new simulations: First, according to the literature composition. However, we also focussed on the Fe the reviewer pointed out in particular. We decreased the iron content by 1.1 at.% points to make it match to the literature value. As SDTrimSP requires the composition to sum to unity, we studied 3 cases:

1. We took the 1.1 at.% and spread them evenly across the remaining components.
2. We added the 1.1 at.% to the abundance of O, the element with the highest concentration.
3. We added the 1.1 at.% entirely to Ca, the heaviest element (with an abundance >0.5 at.%) where one would thus expect the most significant influence on the total sputter yield.

The SDTrimSP simulation results for all cases are pretty much identical and we could not experimentally resolve any difference between them. So even though the deviations from the literature composition could mineralogically describe a slightly different sample, the influence on the sputter yield is negligible. We include this figure along with a brief discussion in the newly created supplementary materials.

On a similar note, SpuBase comes with example compositions for both the lunar highland anorthosite and the mare basalts. These differ in their Ti and Fe content by roughly 3 at.% points and 0.6 at.% points, respectively. Nonetheless, the variation in total sputter yield is still significantly smaller than our experimental uncertainty. See the figure above, which we also include in the supplementary materials. A criterion for negligibility of these discrepancies could thus be in the context of our QCM measurements that the composition variation has to concern elements that are either low enough in abundance or low enough in atomic mass such that they do not contribute significantly to the total removed mass per ion. Exchanging abundant *and* heavy species leads then to the larger differences that are shown for different minerals in Fig. 3 of the manuscript.

In the manuscript, we adapted the Sample Characterization:

The atomic concentration of the main components observed in the samples are presented in Table 1 and agree well mostly with literature [85,86], apart from some slightly off-stoichiometric concentrations

for the heavier elements like Ca, Fe or Ti. The influence of these deviations is discussed in Section III as well as the Supplementary Materials in Supplementary Figs. 2 and 3.

We also added to the Discussion:

Moreover, this is also the reason why the total mass sputter yield is rather robust against slight deviations in sample composition: A variation in abundance of two or three percentage points of a given species will not manifest in resolvable changes of the mass yield. Even though some components in our samples, mostly minor in abundance, might deviate within this range from the literature concentrations (cf. Table 1 in Section IVB), this does not alter the measured sputter yields, because these species contribute little to the total yield. We further elaborate in this point in Supplementary Figs. 2 and 3.

Because this notion is now more thoroughly discussed, we omitted the following sentence from the next paragraph in the Discussion:

~~Moreover, we also varied the composition input to the simulations within the error bars from the sample analysis (Table 1 in Section IVB) and found only an insignificant level of deviation in the results.~~

in figure 3, the used colors in the plots are difficult to distinguish. Light blue is used for different cases, the pink and violet are quite similar, making difficult to discriminate the right curve. The interpolated curves are not just linear interpolations. Are they a polynomial interpolation?

Thank you. We wanted to keep the lines styled as they appeared in previous figures, but we realize that this resulted in a loss of clarity. We therefore changed the color of the SpuBase line to a dark grey and the SDTrimSP line to a teal color. This should make the curves better distinguishable.

We used a cubic spline to interpolate the data points. We have included this information in the figure caption, such that it now contains the following:

Connecting lines between experimental data are interpolated using a monotonic cubic spline to guide the eye.

Reviewer #2

(Remarks to the Author):

The paper reports experimental measurements and modeling estimates of the lunar regolith's sputtering yield under solar wind bombardment, a crucial parameter for quantitatively estimating the weathering rate of the lunar surface, the relative importance of sputtering versus micrometeoritic bombardment to the weathering, the escape rate of energetic (i.e. sputtered) atoms from the surface to space, and the source of material to the lunar exosphere. The study uses Apollo regolith samples to measure/model the yield versus ion incidence angle and morphology (flat vs rough), for 1 keV/amu H and He ions. The experimental measurements give yields substantially lower, by almost an order of magnitude, than model based estimates that do not take into consideration material composition and regolith morphology. The paper shows that more advanced modeling approaches that include these effects produce lower sputtering yields in better agreement with experiment than basic SRIM simulations. However, the updated models still disagree substantially enough with experiment to require additional scaling factors to match the data. The findings suggest that sputter-produced erosion/loss of the surface material, and the source of material to the lunar exosphere, may be substantially below previous estimates. Overall I think the findings are significant, but to assess their actual impact I have a few questions (below, in rough order of importance), which may merit clarification in the manuscript.

We thank the reviewer for their generally positive assessment of our work and do our best to address all remaining questions below. Upon reading the reviewer's comments, we realized that many questions concern a lack of clarity. We added clarifications/elaborations into the manuscript where appropriate and requested.

I am trying to understand what the paper is (or is not) intending to recommend regarding how to apply these data. Presumably the next step might be to use the SDTrimSP model to other energies, so that the full solar wind flux / energy spectrum can be integrated to estimate the Moon's total sputtered flux. To this point, is it justifiable to apply the 0.47 and 0.66 scaling factors to energies other than 1 keV/amu?

From a purely fundamental point of view, we want to convey that getting realistic sputter yields for exosphere modelling purposes requires consideration of realistic surface inputs and ideally experimental benchmarks for the material of interest. From an applied perspective, we provide numbers for both the hydrogen and helium that are applicable for a range of lunar latitudes and robust throughout the expected lunar geology (for this, see also our response to reviewer #1). To emphasise this point more, we elaborated on the closing paragraph of the Introduction:

Our findings provide realistic sputter yield estimates for actual lunar regolith **of $7.3e-3$ atoms/ion and $7.6e-2$ atoms/ion for H and He, respectively**, which are up to an order of magnitude smaller than previous estimates. ~~in line with the conclusions of [44] that micro-meteoroid impacts play a more significant role in lunar space weathering than previously recognized.~~ **These values are largely independent from the ion incidence angle, i.e. the solar zenith angle, and thus valid over a wide range of lunar latitudes. They are furthermore robust against slight variations in sample composition and therefore representative for lunar geology beyond the specific Apollo sample 68501.**

Regarding the scaling factors and the energy dependence: The necessary correction depends primarily on the sample material and the projectile. For instance, the model adaptations proposed in [22] are

shown therein to work for CaSiO_3 (for which they were derived) under 2 keV Ar bombardment as well as for 4 keV ^4He and 3 keV ^3He . Only for protons, further corrections are necessary due to their implantation and chemical sputtering [22]. We would therefore expect our scaling factors to be similarly applicable for a range of energies. On the other hand, this might not even be necessary; Morrissey et al. showed in [34] that when simulating sputtering by ions with energies distributed according to the slow solar wind energy distribution, the resulting yields are only up to 5% lower than for the approximation of 1 keV/amu. The energy is more important for proper descriptions of implantation and depth profiles of ion-induced damages. Of course, experiments to extend our findings to other energies are possible; We cautiously doubt, however, that they would bring novel benefits.

As a general comment, some sort of sentence, maybe at the end of (or beginning of) the introduction that explained exactly (I mean specifically) what was done that is new, would have been extremely helpful to me as the reader. For example, something like: "In this paper, we report new laboratory sputtering yield measurements from 1keV/amu H and He irradiated flat and rough pellet lunar material samples extracted from Apollo 16 sample 68501, and compare against the results of new SRIM, SpuBase, and SDTrimSP models."

We adapted the last paragraph of the introduction outlining the contents of the manuscript according to the suggestion. We also included clarifications about the porous model, in line with following comments. The last newly added sentence will further help clarify points from upcoming remarks.

(...) Our study combines experimental and numerical approaches to quantify sputter yields from actual lunar material (Apollo 16 sample 68501) rather than typically used analog minerals, irradiated with H and He ions at solar wind velocities of ≈ 440 km/s. 1 keV/amu (corresponding to a velocity of ≈ 440 km/s). In particular, we report new laboratory sputter yield measurements for both flat and rough samples prepared from Apollo soil 68501 and irradiated by 1 keV/amu H and He, and compare against the results of SRIM, SpuBase, and SDTrimSP models. Moreover, we model the effect of regolith porosity in addition to rough, but compact, surface morphologies – an effect that is currently not accessible experimentally. We demonstrate that the commonly used simulation codes (...)

Maybe I missed something, but I am confused about the red line in Figure 2. Was an unprocessed Apollo sample (porous regolith) also measured in the lab? The red line appears as though it's connecting points but no points are shown. Section IV.A. only talks about experiments on thin films and pellets, so it was not clear to me what was done with the porous regolith (is this just a model calculation?).

For the regolith case (the red line in Fig. 2), there are unfortunately no experimental data available. In our lab, the ion beam is extracted from the ECR source horizontally, and the samples are mounted on a vertically hanging manipulator. For obvious reasons, this is not possible for loose regolith without some form of handling, compacting and thus altering its properties.

We are therefore constrained to simulations only for this case. However, we are confident that even with the absence of experimental data in this case, the red lines describe physically valid and relevant sputter yields. On the one hand, the actual geometric model in use was already verified in previous studies by comparison to in-situ data from Chandrayaan-1 (references [40, 41] in the manuscript). On the other hand, at this point any material-dependent overestimation of sputter yields by SDTrimSP is already factored out through the application of the correction factors illustrated in Fig. 1. These factors were derived for the 1D case and are not changed anymore once roughness and porosity are introduced. In

that regard, the pellet study forms an important validation case: For the pellet, we have both measured and simulated sputter yields. After the application of the correction factors from the 1D simulations, the experimental and numerical data agree excellently. We can thus conclude that the 3D simulations can accurately model the influence of surface morphology on the sputter yield, once material-dependent shortcomings are addressed.

To address this, we adapted the paragraph concerning sample roughness in the Discussion Section, also partially in response to the remark about surface morphology:

(...) In a further step, Cupak et al. provide a Monte-Carlo-style algorithm called SPRAY that allows allowing the calculation of sputter yields from atomic force microscopy (AFM) images (...). We found excellent agreement to our experimental results using this approach as well, pointing towards surface roughness as the main driver behind the observed sputter yield reduction for the pellet samples. Moreover, these additional simulation results also match with the SDTrimSP-3D data. We are thus confident that the 3D simulations capture the surface structure effects well, once the initial material-dependent overestimation is corrected. We give the SPRAY results and a more in-depth description of the code in Supplementary Fig. 1 and the Supplementary Discussion. We are thus confident that the 3D simulations capture the surface structure effects well, once the initial material-dependent overestimation is corrected.

As SDTrimSP calculates sputter yields always for a given, but fixed, incidence angle, the regolith data are available for a discrete set of incidence angles (0° , 15° , 30° , 45° , 60° , 75° , 80° , 85°). As the porous 3D calculations are computationally more expensive than the much simpler 1D ones, the distance between angles is larger and only decreased for the grazing incidence regime. This leads to the look the reviewer mentioned. We decided to present the data with connecting lines to keep the same convention throughout the manuscript (lines for simulations, markers for experimental data).

To make the numerical nature of the regolith data clearer, we edited the corresponding paragraph in the Results section:

As sputter yield measurements for loose regolith powder are not feasible in our setup, we introduced porosity in SDTrimSP-3D calculations (red lines). In this case, ~~When porosity is introduced through the implementation of regolith structures in SDTrimSP-3D (red lines),~~ the same effect is observed (...)

It's hard for me to assess the paper's claim that morphology is an important factor for the yield, since Fig. 2 seems to show only a minor difference between the flat and rough pellet cases, and (again) it's unclear to me what the regolith case is supposed to be in Fig 2 (my previous question).

The concept that the surface structure influences the sputtering behaviour is not new and has been discussed for decades. See, for example, the following work by Küstner: M. Küstner et al., Nucl. Instrum. Methods Phys. Res. B 145 (1998) 320–331, doi:10.1016/S0168-583X(98)00399-1. While the data presented in our manuscript might not look too different between the sets from the flat and the rough samples, it is important to point out that we could only study one particular roughness, as determined by the pellet pressing process. A systematic study covering various degrees of roughness (expressed both in terms of the RMS roughness as well as the surface inclination distribution) is given by Cupak et al. in reference [69] of our manuscript. A supplementary analytical description can be found in [70]. From a fundamental physics perspective, the difference between the flat and rough sample sputter yields is

indeed not “minor”: Note that for both projectiles in Fig. 2, there is about a factor of 2 difference between the sample configurations under grazing incidence without overlapping uncertainties. For technical or commercial applications of sputtering, such as in preparing atomistically clean surfaces or sputter coating of materials, such a factor is very noticeable rather than negligible.

As outlined also in our response to the previous question, the yield reduction for the regolith case also comes from the surface structure of the sample, and we are confident that the simulation accurately models the effect of surface morphology. To further untangle the morphology influence from the BCA simulations, we employed the SPRAY code provided by and described in ref. [69]. We included this argument in the Discussion Section as well, but reiterate the main points here to clear up confusions: The SPRAY codes maps 1D simulations onto real surfaces (AFM images of the pellets in our case) and does not calculate the whole collision cascade again. If the 1D input is accurate (cf. Fig. 1 in the manuscript), any sputter yield deviations have to be from surface effects. To illustrate this point more comprehensibly, we added a figure and a description of the SPRAY code to the newly created supplementary materials. We also added a reference to the supplementary materials in the Discussion Section about the roughness influence.

We believe that the origin and meaning of the regolith line is now evident from our description in the last comment (above).

It's not explained what a "pellet" actually is, how it's prepared, etc. I understand that it's explained in reference 29, but considering that pellets are one of the main cases studied, does it not merit at least a sentence or two of explanation in the paper?

We changed the beginning of Section IIB:

A comparison of the total sputter yields for flat PLD films, rough pellets, and porous regolith samples is given in Fig. 2A and Fig. 2B for 1 keV/amu H and He, respectively. Just like in the previous subsection, experimental data for flat samples were obtained from PLD films. For measurements on rough surfaces, we pressed some of the original regolith sample in order to form stable pellets that can withstand mounting in the vacuum vessel. Experimental data for these were obtained by means of a catcher QCM (cf. Section IVA). As this is not possible for the porous regolith case, only simulated data are available for this instance. For preparation and characterization, the reader is referred to Section IVB. Note that for (...)

We also elaborated on the pellet production process in Section IVB:

All samples used in this study were produced from Apollo soil 68051. This material is a mature (Is/FeO = 85) specimen collected during the Apollo 16 mission with an agglutinate content of about 38% and average grain sizes of $\approx 100 \mu\text{m}$ [85]. To study the influence of surface morphology, samples with rough surfaces were prepared in the form of pellets pressed from the lunar regolith. For this purpose, we used circular stainless-steel holders into which a layer of KBr was pressed to increase the cohesion between the sample and the back plate. Onto this interlayer we then pressed the lunar sample. The pellet preparation process is described in greater detail in [29], including the press specifications and illustrative photos of the pellets, both in press and after successful preparation. However, in contrast to the analogue mineral pellets described therein, we did not filter by grain size to keep the sample as representative as possible. Pellets were pressed following the procedure in [29], but without filtering for grain sizes.

A layer of KBr was first pressed into the holder to increase cohesion between the regolith and the back-plate.

One such regolith pellet was subsequently used as a donor in pulsed laser deposition to grow flat thin films on QCM substrates. Thin films were deposited onto QCMs by means of pulsed laser deposition. This was done The depositions were performed under (...)

Figures 1 and 2: How are we to interpret the y-axes? Why are there two scales?

Both y-axes give sputter yields. In both figures and all subpanels therein, the left axis gives the sputter yield in atomic mass units per incident ion. This is the physical quantity that is directly accessible by the quartz crystal microbalance technique and therefore the most accurate representation of our measurements. On the other hand, BCA simulations typically give results on an atomistic basis, i.e. sputter yields will be given in ejected atoms per ion. This is usually the preferred unit also for exosphere modellers, which is why we opted to give both units.

Calculating the yields in amu/ion from atoms/ion is rather straightforward and follows from multiplication with the species' atomic mass and addition of all partial, species-specific sputter yields. The other direction is a little bit more involved; Due to the preferential sputtering of more volatile elements in compound materials, sputtering will dynamically change the surface composition until a steady state is reached. In this steady state, the surface composition will be different compared to the bulk composition, but the sputtered particle fluxes will follow the bulk stoichiometry. Given the total sputter yield and the bulk composition, one can thus also calculate a sputter yield in atoms/ion from our QCM measurements. The necessity of a fixed composition in this calculation is also the reason why in Figure 3, where different minerals are compared, we refrained from including a second y-axis. We elaborated on this point in the manuscript and added reference [45]. The corresponding paragraph in the Results Section now reads:

The sputter yields are given as function of incidence angle α in atomic mass units per incident ion (left axis of both panels) and atoms per incident ion under the assumption of stoichiometric particle fluxes (right axis of both panels). While our QCM technique (see Section IV A) directly measures the sputter yields as mass changes (left axes), simulations typically give them in atoms per ion. Sputtered atoms per ion is also usually the unit of choice for exosphere models (see, e.g., [12]), which is why we give both units wherever possible. Conversion of the experimental data to atoms per ion was carried out under the assumption that in the steady state, the elemental sputtered particle fluxes correspond to the bulk stoichiometry [22, 45].

We do realise, however, that the left label in Figure 1 mistakenly reads "Sputter Yield Y (atoms/ion)" where the unit should have been amu/ion. We corrected the axis label and thank the reviewer for their keen observation.

The thin film is described as flat (line 262), but why does the AFM image in Fig. 4 not appear flat? Presumably I am misinterpreting it, but it appears to me to have nearly zero thickness (according to the color scale) with a few discrete particles.

The thin film is indeed flat with some discrete particles that are a product of the pulsed laser deposition process. The 0-level of the colour code was chosen to represent the point of lowest elevation in the AFM image, i.e., the surface level of the film. Rather than a film of near-zero thickness, the image should thus be interpreted as dense, flat film with near-zero elevations from its surface; the exception to this being

the few particles. From the AFM images we also extracted the distribution of surface inclination angles with respect to the surface normal. For a perfectly flat surface, this distribution would have the shape of a delta-distribution centred at 0° . Our flat films have a mean located at 6.2° (as indicated in the figure), making them “flatter” by this metric compared to films in the conceptionally similar studies of Cupak et al. and Biber et al. (references [69] and [24] in the original manuscript, respectively).

We also know that the films are finitely thick and dense, coating the entire sensitive quartz area, rather than having nearly zero thickness. On the one hand, we have experience in our lab with dealing with such mineral films and their characterisation (see, i.e., references [22, 24, 28, 56] in the manuscript, but most importantly [23]). On the other hand, a partially exposed gold electrode of the QCM (the substrate of the films) would exhibit a much higher sputter yield due to high atomic mass of Au. However, we only see sputter yields increasing after applying sufficient fluence to erode the tens of nm of sample film.

To make things clearer in the paper, we added to the following sentence:

The film sample is generally flat **as indicated by the uniformly colored surface base level** with the exception of some particles that formed during the PLD deposition.

Figure 3. Should it not be clearly explained in the caption that this is showing the flat surface case?

We agree and include the following sentence with the figure caption:

For all cases, data are compared for flat samples.

Section IV.A. It's explained that a QCM is used in the thin film case to measure the sputtering yield. Is this also done for the rough pellet case? Is it not more difficult to get the QCM to resonate with crushed material (as compared to a PLD deposited film)? Or is only the catcher QCM being used for the pellet case? The paper was not very clear on this.

For the rough pellet case, the catcher QCM technique was used as an extension to the direct one. With the catcher configuration, we probe the angular distribution of the sputtered ejecta as a slice through the emission plume in the polar angle direction. However, this information alone is not sufficient to quantitatively reconstruct the sputter yields of a rough target: We experienced in the past that for compound targets, not every species necessarily has the same sticking probability. Therefore, integration over all polar angles of the measured distribution does not uniquely identify the rough sample sputter yields. We deal with this problem by alternating irradiations of rough samples and the thin films on QCM targets. For the latter, the sputter yields are directly measured, simultaneously to the angular distribution. A known sputter yield thus gives a known angular distribution. Therefore, the rough sample sputter yield is then determined by the flat sample sputter yield, scaled with the ratio of the integrated angular distributions of both sample types. This more involved experiment is the also the reason why measuring sputter yields of rough samples takes a lot longer, and consequently, we present fewer data points for the pellet case in Fig. 2.

More concisely, no crushed material was used directly on a QCM and the catcher QCM method had to be employed. Our research group started to develop this technique as early as 2017 (cf. [77] in the paper) with gold targets as a proof of principle and applied it to rough mono-elemental W targets [69] and finally minerals [24]. It is thus well-documented and reliable. An additional approach by another group with catcher QCMs covering also the azimuthal direction and a discussion about sticking

coefficients can be found in the following publication: C. Bu et al., J. Appl. Phys. 135 (2024) 035302, doi:10.1063/5.0184417.

To make it clearer that for the pellets, we used the catcher technique, we added a sentence to first appearance of these data in the Results section (see also our response to the pellet question above).

Minor edits: Line 189: “allowing to calculate” should be “allowing the calculation of”. Also line 273: “allows to probe” should be something like “allows the angular distribution of the sputtered ejecta to be probed”.

We rephrased these parts according to the reviewer’s suggestions.